# Research Progress on Micro(nano)plastic-Induced Programmed Cell Death Associated with Disease Risks

**DOI:** 10.3390/toxics12070493

**Published:** 2024-07-05

**Authors:** Huanpeng Liu, Huiqi Li, Ting Chen, Fan Yu, Qizhuan Lin, Haiyang Zhao, Libo Jin, Renyi Peng

**Affiliations:** Institute of Life Sciences & Biomedicine Collaborative Innovation Center of Zhejiang Province, College of Life and Environmental Science, Wenzhou University, Wenzhou 325035, China; 22461338015@stu.wzu.edu.cn (H.L.); 22461338011@stu.wzu.edu.cn (H.L.); 23461044002@stu.wzu.edu.cn (T.C.); 23461044029@stu.wzu.edu.cn (F.Y.); 23461044010@stu.wzu.edu.cn (Q.L.); 20160106@wzu.edu.cn (H.Z.); 20160121@wzu.edu.cn (L.J.)

**Keywords:** MNPs, programmed cell death, diseases, targeted therapy strategies

## Abstract

Due to their robust migration capabilities, slow degradation, and propensity for adsorbing environmental pollutants, micro(nano)plastics (MNPs) are pervasive across diverse ecosystems. They infiltrate various organisms within different food chains through multiple pathways including inhalation and dermal contact, and pose a significant environmental challenge in the 21st century. Research indicates that MNPs pose health threats to a broad range of organisms, including humans. Currently, extensive detection data and studies using experimental animals and in vitro cell culture indicate that MNPs can trigger various forms of programmed cell death (PCD) and can induce various diseases. This review provides a comprehensive and systematic analysis of different MNP-induced PCD processes, including pyroptosis, ferroptosis, autophagy, necroptosis, and apoptosis, based on recent research findings and focuses on elucidating the links between PCD and diseases. Additionally, targeted therapeutic interventions for these diseases are described. This review provides original insights into the opportunities and challenges posed by current research findings. This review evaluates ways to mitigate various diseases resulting from cell death patterns. Moreover, this paper enhances the understanding of the biohazards associated with MNPs by providing a systematic reference for subsequent toxicological research and health risk mitigation efforts.

## 1. Introduction

Microplastics (MPs) are plastic fragments and particles with diameters less than 5 mm [1,2], and particles smaller than 1000 nm are classified as nanoplastics (NPs) [3]. MNPs are widely distributed in many ecosystems, including oceans, lakes, soil, and farmland [4,5]. However, MNPs are not confined to specific ecosystems; they can be found virtually anywhere in the world, influenced by numerous complex migration pathways such as wind currents, water currents, and biological activities [6,7]. For the past few years, increasing attention has been focused on the adverse impacts of MNPs on various organisms. Reports indicate that the toxicity of MNPs to various organisms is predominantly sublethal [8,9], and their toxic effects can be categorized into three primary forms: including the release of toxic chemical components, acting as carriers to adsorb harmful substances, and causing physical damage to biological organs through inhalation or ingestion [10,11]. Human activities have led to the widespread pollution of ecosystems with MNPs, resulting in inevitable human exposure to MNPs [12]. Bharath K et al. examined the type of MNPs in groundwater and found that the main components were polystyrene (PS) and polypropylene (PP), while small amounts of polyethylene (PE) and polyvinyl chloride (PVC) were also found [13]. Given that humans occupy the top of the food chain, they unavoidably consume food items, such as water and seafood, that are contaminated by MNPs in their daily lives. Additionally, humans might encounter MNPs through inhaling airborne particles and skin contact. Zhu et al. performed a systematic analysis of MPs larger than 20 μm in the human digestive and respiratory systems, and revealed their ubiquity in the human body [14]. Numerous studies have examined the detrimental effects of MNPs on the human body [15,16,17,18,19]. Lin et al. conducted a study on the impact of 80 nm PS-NPs on mitochondrial metabolic pathways and function in normal liver (L02) cells and lung (BEAS-2B) cells. They showed that exposure to NPs could not result in significant cell death; instead, it induced mitochondrial dysfunction [20]. Wu et al. assessed the low toxicity of MPs by examining alterations in cell survival, intracellular reactive oxygen species (ROS) levels, membrane integrity and fluidity, and mitochondrial depolarization in Caco-2 cells exposed to PS-MPs [21]. Weber et al. demonstrated that NPs induced inflammation in monocyte-derived dendritic cells and primary human monocytes [22]. Workers in the plastics and textile industries are chronically exposed to plastic fibers and have been reported to experience microplastic-induced lung damage, which is characterized by inflammation, allergies, and fibrosis [23,24]. Pulmonary inflammation is closely associated with disturbances in various modes of cell death, including pyroptosis, autophagy, and apoptosis [25,26,27,28,29]. Halimu et al. reported that PS-NPs induced apoptosis in the human lung epithelial cell A549 [30], Wang et al. reported that nanoparticles formed by silica induced inflammation by causing lysosome damage and dysfunction in autophagy through ROS/PARP/TRPM2 signaling pathways. [31], and Wu et al. reported that exposure to amine-polystyrene (APS) -NPs activated NLPR3 inflammasomes in MLE-12 cells, thereby inducing pyroptosis [32]. A recent study revealed that anionic NPs enhanced the aggregation of Parkinson’s disease-associated α-synuclein, thereby exacerbating the spread of α-synuclein pathology in interconnected vulnerable regions of the brain [33]. In summary, extensive exposure to MNPs can induce various types of cytotoxicity [34,35], and increase the risk of disease development. Marfella et al. were the first to demonstrate a link between MNPs and human health. Their research showed that MNPs infiltrate human arteries, significantly increasing the risk of serious diseases, including heart disease, stroke, and mortality [36]. However, the size and concentration of toxic MNPs vary among different species, influenced by the individual organism’s response to external stressors. A new study indicated that patterns of MPs in bronchoalveolar lavage fluid (BALF) in children with lung disease [37]. Consequently, MNP exposure is closely linked to human health and may result in damage to the reproductive system, nervous system, cardiovascular system, respiratory system, digestive system, immune system, endocrine system, and motor system [38,39] (Appendix A).

Numerous studies have shown that MNPs can induce varying degrees of toxicity in numerous species, including mice, zebrafish, oysters, chickens, earthworms, nematodes and sea urchins [40,41,42,43,44,45,46]. This toxicity induces liver toxicity, neurotoxicity, reproductive toxicity, and developmental toxicity. Li et al. demonstrated that PS-MPs could disrupt glycolytic flux and induce apoptosis through calcium overload in a mouse model [47]. Another study demonstrated that stress responses caused by exposure to PS-MPs exacerbated lipopolysaccharide (LPS)/d-galactosamine (d-GalN)-induced death in mice and increased the risk of liver damage [48]. In toxicology research, biological response pathways can be used to explain the pathway by which toxicity arises and how it ultimately leads to adverse health consequences for the organism. Thus, monitoring pre-disease events at the cellular level can effectively predict the harm caused by toxic substances to organisms. The physiological state of cells can reflect cellular responses to external substances [49]. Hence, it is crucial to assess the virulence of environmental exposure to MPs and the correlated metabolic pathways at the cellular level. A previous review delineated PCD induced by MNPs and scrutinized the various factors contributing to distinct forms of PCD triggered by MNPs, including particle size, functional group modification, surface potential, aging, and co-exposure. The particle size of MNPs, differences in surface-modified charge, aging, and co-exposure with other substances often influence changes in the mode of PCD [50]. Similarly, different exposure models produce varying toxic effects. Model animals, such as mice, zebrafish, and chickens, as well as specific cell lines, are often used to investigate MNP-induced PCD. To understand the potential biological impact of MNPs, we collected nearly five years of research papers on PCD caused by MNPs (Table 1) and were surprised to find that only two studies used PP-MPs, one study used PE-NPs, while the other studies used PS-MNPs. Additionally, long-term exposure at low doses is usually studied in animals, whereas acute exposure at higher doses is typically examined in cells. This review offers a comprehensive examination of microplastic-induced pyroptosis, ferroptosis, autophagy, necroptosis, and apoptosis, and sheds light on the pathways and mechanisms of MNP-induced cytotoxicity. Indeed, the various forms of PCD induced by MNPs are closely intertwined. Variations in environmental conditions, MNPs’ type and size, and species differences can prompt reciprocal conversions of PCD. Additionally, it summarizes the prospects and challenges of these processes in human diseases.

## 2. Research Status of PCD Caused by MNPs and the Association between PCD and Disease

This review provides a systematic analysis of the PCD induced by MNPs along with their close association with disease. Following the identification of MNP-PCD links, we conducted a comprehensive manual sea rch on Web of Science using predefined keywords related to MNPs and cell death, including (“microplastics” or “nanoplastics”) and (“pyroptosis”, “ferroptosis”, “autophagy”, “necroptosis”, or “apoptosis”). Additionally, we employed predefined keywords associated with cell death and disease, such as (“pyroptosis”, “ferroptosis”, “autophagy”, “necroptosis”, “apoptosis”, or “cuproptosis”) and (“disease”). Our database searches were restricted to articles published within the last decade.

We employed VOSviewer software (Version 1.6.20) to gain insights into these articles, conducting an analysis of terms, keywords, and categories. Prominent keywords shed light on PCD induced by MNPs and underscore the strong association between PCD and disease. This correlation not only delineates a significant avenue for research but also aids in organizing and summarizing subsequent findings. Initially, 423 papers were preliminarily identified over the past decade. Among them, there are 320 studies on MNPs, of which 121 studies focus on the generation and analysis of MNPs and pay little attention to the biological effects of MNPs. After checking the title, abstract, keywords, and abbreviations and considering their impact, 241 papers were included in this review. Utilizing VOSviewer, bibliometric analysis encompassed titles, abstracts, and keywords from 2015 to 2024. Additionally, we performed a keyword evolution analysis to pinpoint key research areas and identify recent trends (Appendix A). The analysis revealed that 62 keywords were cited at least 15 times each. Essential keywords such as “MNP”, “cell death”, and “disease” have emerged as focal points in current research, underscoring the imperative for continued investigation in these areas.

## 3. Molecular Regulatory Mechanisms of Pyroptosis Induced by MNPs and Associated Disease Risks 

### 3.1. Pyroptosis Caused by Micro(nano)plastics

Pyroptosis represents a form of PCD facilitated by Gasdermin-D (GSDMD), typified by cellular swelling culminating in membrane rupture, thereby releasing cellular contents and eliciting a robust inflammatory reaction [86]. GSDMD is both the substrate of caspases-11/4/5 and the central executor of pyroptosis. The pore-forming activity of the Gasdermin-N domain drives pyroptosis, resulting in cell death [87,88,89]. Numerous studies have demonstrated that exposure to microplastics induces pyroptosis in cells in various organisms, including mice, chickens, and humans. Wei et al. demonstrated myocardial toxicity in mice exposed to PS-MPs and revealed that MPs induced pyroptosis through the GSDMD/Caspase-1/NLRP3 pathway [52]. Similarly, Hou et al. demonstrated that exposure to PS-MPs triggered pyroptosis in rat ovarian granulosa cells via activation of the NLRP3/Caspase-1 pathway [53]. Zhang et al. investigated the effects of various concentrations of 5 μm PS-MPs on chicken primary cardiomyocytes and discovered that PS-MPs induced pyroptosis via the NF-κB-NLRP3-GSDMD pathway (Appendix A) [54]. Zhong et al. established a mouse model of exposure to arsenic (As)-polystyrene microplastics and found that combined exposure to MPs and As triggered pyroptosis in liver cells via the Caspase-1/NLPR3 pathway by targeting the expression of ASC, NLPR3, GSDMD, and Caspase-1 [90]. Importantly, several studies have reported that PS-MPs could induce toxicity in human (micro) vascular sites, which was characterized by hemolysis, thrombosis, coagulation, and damage to vascular endothelial cells. The primary causes of these effects include oxidative stress, apoptosis, and pyroptosis [91,92]. These findings suggest that MPs are a potential cause of some cardiovascular diseases, such as myocarditis, atherosclerosis, and myocardial infarction.

### 3.2. Regulatory Mechanism

Exposure to MNPs can result in the excessive accumulation of ROS within organisms and imbalance between oxidation and antioxidant effects, consequently inducing oxidative stress [93,94]. The toxic impact of MNPs on organisms in vivo and in vitro are mediated by the induction of oxidative stress [93,95]. When cells exposed to MPs encounter oxidative stress, NF-κB can undergo [96,97]. Phosphorylated NF-κB plays a role in activating the NLRP3 inflammasome [97]. Furthermore, there is evidence to suggest that MNPs can trigger the inflammatory NF-κB/MyD88/NLRP3 pathway [98]. The close relationship between oxidative stress and proinflammatory processes has been acknowledged for an extended period. A number of studies have proved that oxidative stress can directly trigger inflammation, thereby initiating inflammatory responses and the release of chemokines [99]. The intracellular pattern recognition receptor NLRP3 forms oligomers and exposes its clustered pyrrole ring domain, facilitating signal recognition. It then binds to the precursor of Caspase-1 via the adaptor protein ASC, forming a multiprotein complex that activates Caspase-1 [100]. Activated Caspase-1 performs dual roles: First, it cleaves GSDMD, leading to the formation of peptides containing the active domain of GSDM-NT [101,102]. This induces cell membrane perforation, disrupts osmotic potential, causes cell rupture and content release, and triggers an inflammatory response [103,104]. The Gasdermin-N domain of GSDMD alone is adequate to induce pyroptosis in any cell [105]. Second, the IL-18 and IL-1β precursors are cleaved by activated Caspase-1, generating active IL-18 and IL-1β, and then released from the cell [106]. This release recruits inflammatory cells, amplifying the inflammatory response (Figure 1). However, in certain conditions, macrophages, dendritic cells, and neutrophils may withstand lysis induced by inflammasome-activated GSDMD and do not undergo membrane rupture or pyroptosis, which is termed hyperactivation [86]. Given that hyperactivated cells can release increased amounts of inflammatory mediators, it is feasible to differentiate between hyperactivation and pyroptosis by assessing lactate dehydrogenase levels in the cell culture supernatant [107]. However, the precise mechanism that determines whether GSDMD cleavage results in pyroptosis or hyperactivation remains unclear. NPs can activate various signaling pathways, including the NLRP3, Caspase-1, and NF-κB pathways, via oxidative stress and inflammation [32,90]. This activation significantly upregulates the expression of IL-18 and IL-1β, facilitates the cleavage of GSDMD, induces pyroptosis, and ultimately results in tissue damage in vivo. Hence, pyroptosis may serve as the cellular mechanism through which MPs exert adverse effects on organisms via ROS production and NF-κB activation. Targeting inflammation-dependent pyroptosis could be a novel strategy for mitigating the adverse effects of MPs. Studies have shown that ASC can augment the recruitment and phosphorylation of Caspase-1, and treatments targeting ASC can effectively decrease the activation of Caspase-1 and IL-1β, consequently reducing tissue damage [108,109].

### 3.3. Therapeutic Strategies Targeting Pyroptosis in Cancer, Neurodegenerative Diseases and Immune Diseases

Pyroptosis represents a critical innate immune response, serving a pivotal role in responding to endogenous danger signals and counteracting infections [110]. It is extensively implicated in the onset and progression of immune disorders, nervous ailments, and infectious diseases [111,112,113,114]. Despite existing knowledge gaps regarding the precise involvement of pyroptosis in diverse pathological conditions, conducting thorough investigations into its signaling pathways, regulatory mechanisms, and pathological implications holds immense potential for the development of novel therapeutic strategies aimed at preventing and treating a wide array of human diseases.

GSDMD knockdown has been shown to activate Caspase-3 cleavage in lung cancer cells of mouse, inducing the inhibition of tumor growth in mice [88]. Currently, numerous GSDMD inhibitors including disulfiram, necrosulfonamide, dimethyl fumarate, LDC7559, and angpomegranate have been identified [115]. During dendritic cell treatment, the absence of NLRP3 has been shown to significantly decrease melanoma metastasis in the lungs [115]. N-acetylcysteine (NAC), which is a common antioxidant, inhibits ROS production, suggesting its potential for preventing NLRP3 inflammasome activation [116]. These findings indicate that inhibiting GSDMD and the inflammasome may be a promising strategy for tumor treatment. Moreover, various components of the NLRP3 inflammasome, such as Caspase-1, NLRP3 and ASC, are markedly upregulated in microglia within the substantia nigra of Parkinson’s disease patients, suggesting a link between pyroptosis and Parkinson’s disease [117]. Compounds such as flufenac and mefenac have been shown to inhibit NLRP3 activation, thereby alleviating the symptoms of Parkinson’s disease [118]. Therefore, targeting the NLRP3 inflammasome and its associated molecules could be a therapeutic strategy for neuroinflammatory diseases, including Parkinson’s disease. The gradual depletion of CD4 T cells in individuals infected with HIV is a primary factor that contributes to AIDS [119]. Caspase-1-mediated pyroptosis occurs when CD4 T cells fail to infect with the virus. However, caspase-1 inhibitors such as VX-765 and AC-YVAD-CMK may alter this process [120,121], suggesting the possibility of a novel therapy that targets the host rather than the virus to treat AIDS [122]. In summary, targeting GSDMD, NLRP3, Caspase-1, and related molecules is a promising approach for regulating PCD. However, significant advancements are needed to translate these therapeutic methods into clinical practice.

## 4. Molecular Regulatory Mechanisms of Ferroptosis Caused by MNPs and Associated Disease Risks 

### 4.1. Ferroptosis Caused by Micro(nano)plastics

Ferroptosis, a recently identified type of iron-dependent PCD, is characterized by pronounced lipid peroxidation and iron accumulation. Ferroptosis inducers can indirectly or directly impact glutathione peroxidase, leading to a reduction in the cellular antioxidant capacity and the accumulation of lipid ROS, ultimately resulting in oxidative cell death [123,124]. Mu et al. demonstrated that PS-MPs induced lipid peroxidation in the livers of mice, which upregulated transferrin receptor (TFRC) expression and downregulated ferritin heavy chain 1 (FTH1), the XCT system, glutathione peroxidase 4 (Gpx4), and ACSL4, thereby inducing ferroptosis [51]. Tang et al. observed that exposing mice during pregnancy to PS-NPs caused ferroptosis in the small intestine cells of their offspring [57]. Chen et al. showed that PS-NPs disrupted lipid peroxidation and iron metabolism in zebrafish larvae, thereby inducing ferroptosis [80]. Yin et al. reported that chickens exposed to PS-MPs exhibited increases in hepatic glutamine and glutamate synthesis, which promoted autophagy-dependent ferroptosis via the liver–brain axis [56]. Carbo et al. observed ferroptosis in microalgae exposed to bisphenol A (BPA), which is a major component of MPs, due to the disruption of iron dynamics and reoxidation balance. The reversal of microalgal growth by the classic ferroptosis inhibitor Fer-1 further confirmed that microplastics induced ferroptosis [125]. Moreover, studies have indicated that combined exposure to microplastics and other substances can induce ferroptosis in cells, such as PE-imidacloprid-induced ferroptosis in earthworm cells and PE/PS-polybrominated diphenyl ether-induced alterations in ferroptosis pathways factors in grouper [126,127]. Importantly, excessive cell death due to ferroptosis is related to the pathogenesis of numerous diseases, suggesting that MPs may be potential pathogenic factors associated with disease development.

### 4.2. Regulatory Mechanisms

Exposure to MNPs can trigger ROS accumulation in organisms, and ROS-mediated lipid peroxidation is a primary driver of ferroptosis [128,129]. Lipid peroxidation in cells involves two main processes, one of which is catalyzed by fatty acid enzymes [130]. Polyunsaturated fatty acids (PUFAs) can be converted into highly reactive lipid peroxides by a series of enzymes. Research indicates that in addition to the cellular membrane systems, the primary sources of these PUFAs are arachidonic acid (AA) and linoleic acid, which are widely present within cells [131]. Experimental evidence has confirmed that the addition of AA and linoleic acid during cell culture can expedite ferroptosis [132]. Mu et al. demonstrated that exposure to microplastics induced the expression of ACSL4, which promoted lipid synthesis and subsequent lipid peroxidation [51]. The oxidation of AA is primarily regulated by three enzymes: AA is converted to AA-CoA by acyl CoA synthetase long-chain protein 4 (ACSL4) and then esterified with phosphatidylcholine to form AA-PE by LPCAT3, followed by lipid peroxidation by the lipoxygenase protein family (LOXs) [133]. These enzymes, particularly ACSL4, play critical roles in ferroptosis, and the LOX family members ALOX5 and ALOX12 are targeted by various ferroptosis inducers [133]. Additionally, exposure to MNPs increases the activity of the XC system and GSH system and downregulates Gpx4 expression [134,135]. The main function of the XC system is to transport cystine and glutamate into and out of cells, and while cysteine is involved in the synthesis of GSH via the XC system [136]. Gpx4 acts as a key intracellular lipid peroxidation reducer that regulates ferroptosis by controlling lipid peroxidation. Gpx4 converts GSH to GSSG, thereby inhibiting ROS accumulation [137]. Several studies have reported decreased Gpx4 expression in response to microplastics, suggesting that reduced Gpx4 levels may inhibit the conversion of lipid hydrogen peroxide to lipid alcohol, thereby promoting lipid peroxidation [138,139].

An additional intracellular route for lipid peroxidation involves the Fenton reaction, which is triggered by free iron ions. Iron within the cell is primarily transported by transporters containing trivalent iron, which enter the cell via the transferrin receptor TFR on the cytomembrane [140]. Sun et al. demonstrated that exposure of BV2 cells to microplastics led to a dose-dependent increase in intracellular ferritin levels and TFRC protein expression [55]. Moreover, the pH in the cell is acidic, and free trivalent iron is reduced to bivalent iron by the iron reductase STEAP3 [141]. Iron ions within the cell react with peroxides to generate iron ions and peroxy radicals. These peroxy radicals then attack lipid molecules, oxidizing them into lipid peroxides (Figure 2). Under normal conditions, lipid peroxides are maintained in homeostasis because the concentration of iron ions remains stable. However, a sudden increase in intracellular iron ions significantly intensifies the Fenton reaction, resulting in the excessive accumulation of lipid peroxides, thereby inducing ferroptosis [142]. This process is characterized by reduced mitochondrial mass, increased membrane density, and decreased ridges [143].

### 4.3. The Role of Ferroptosis in Targeted Treatments for Cancer, IRI, and Inflammatory Enteritis

The role of ferroptosis in human diseases has been extensively investigated (Table 2), and researchers have examined treatments targeting ferroptosis-induced pathologies. Notably, cancer is a primary focus due to the ubiquitous expression of Gpx4, which is crucial for cancer cell survival [144]. Additionally, studies have shown that inhibiting the XC system markedly suppresses tumor growth and metastasis in murine models [145]. The XC system functions as a reverse transporter to facilitate the uptake of cystine, a vital component of glutathione (GSH) that counteracts oxidative stress induced by oxidants such as hydrogen peroxide [146]. Ferroptosis has also been implicated in neuropathic disorders. Dixon et al. reported that ferrostatin 1 effectively mitigated glutamate-induced cell death and ferroptosis in cancer cells in mouse brain slices cultured in vitro [147]. Furthermore, Gao et al. provided evidence linking ferroptosis to ischemia reperfusion-induced injury (IRI) through in vitro experiments with mouse embryonic fibroblasts [148]. Given their pivotal role in glutamine metabolism in ferroptosis, these enzymes are potential targets for IRI treatment. Moreover, ferroptosis is closely associated with Crohn’s disease [149]. ROS have been implicated in enteritis pathogenesis, and oral iron supplementation can exacerbate inflammatory bowel disease. Conversely, the iron chelator desferriamine significantly ameliorates IBD symptoms [150]. Chen et al. demonstrated that ferrostatin 1 could alleviate DSS-induced Crohn’s disease in rats by inhibiting the HO-1/Nrf2 signaling pathway [151]. The modulation of ferroptosis to inhibit the onset and progression of diseases has become a prominent area of research and a therapeutic focus, although the functional alterations and specific molecular mechanisms warrant further examination.

## 5. Molecular Mechanisms of Autophagy Induced by MNPs and Associated Disease Risks 

### 5.1. Autophagy Caused by Micro(nano)plastics

Autophagy, an evolutionarily conserved process, is responsible for intracellular material turnover in eukaryotes, serving as a mechanism of PCD. During autophagy, damaged organelles and proteins are sequestered within autophagic vesicles featuring bilayer membranes, and subsequently transported to vacuoles or lysosomes for degradation and recycling [161,162]. A number of researchers have documented the induction of autophagy by MNPs in organisms. Shaoyong et al. administered environmental concentrations of PS-MPs to mice via gavage, and revealed that PS-MPs induced autophagy in mouse cells and facilitated the accumulation of autophagic vacuoles [58]. Lin et al. showed that exposure to PS-NPs activated the AMPK/mTOR/ULK1 signaling pathway in mice, and exacerbated lipopolysaccharide (LPS)-induced autophagy [163]. Nie et al. exposed SH-5Y5Y cells to PS-NPs, and observed an increase in the number of autophagosomes. Western blot analysis confirmed increased expression of LC3B-II/LC3B-I, ATG7, and ATG5, indicating that MPs activated autophagy in SH-5Y5Y cells [59]. Similarly, Ding et al. noted a noteworthy rise in the number of autophagosomes and autolysosomes in GES-1 cells treated with PS-NPs, indicating NPs activated autophagy [62]. A study examined the autophagy effect of PS-NPs with different particle sizes, and small NPs showed a higher likelihood to induce autophagy and autophagosomes formations [60]. Conversely, a study revealed that exposing human nasal epithelial cells to 50 nm PS-NPs reduced autolysosomes [61], suggesting impaired autophagic flux, that was due to severe damage and ultimately led to cell death.

### 5.2. Regulatory Mechanism

Autophagy is a crucial process for intracellular material transfer in eukaryotes that is evolutionarily conserved. Nevertheless, the role of autophagy extends beyond mere substance elimination; it is a dynamic process that generates energy and new building blocks, contributing to homeostasis and cell renewal [164,165]. Autophagy can be classified into three main types: macroautophagy, microautophagy, and chaperone-mediated autophagy (Figure 3), based on distinct substances and transport mechanisms [166].

Several studies have demonstrated that smaller NPs can trigger autophagy in cells (Appendix A). High levels of autophagy can indicate that cells are trying to survive and that death occurs only when this effort fails [167]. Cells exposed to NPs typically internalize them via vesicle-mediated endocytosis, leading to the formation of autophagosomes, which is a critical aspect of subsequent cell death [168]. Given the intimate relationship between autophagy and apoptosis, excessive autophagic activation can culminate in apoptosis, which shares common regulatory signaling pathways governing cell survival and growth. Recent studies have revealed that autophagosomes can induce apoptosis by serving for a platform for caspase activation, particularly caspase-8 [169,170]. In response to NP exposure, autophagy serves as a specific target of the negative regulator of caspase activity through degradation, inducing the increase in caspase activity and thus promoting cell death.

Although autophagy has been traditionally considered as a mechanism of cell death, the current consensus is that its primary role is protective, and it is crucial for maintaining intracellular homeostasis. However, severe damage can disrupt autophagic flux, leading to inadequate autophagy [171]. Zhou et al. used earthworms and observed the internalization of nanoplastics by these organisms. The nanoplastics predominantly accumulated in lysosomes, causing lysosomal swelling and an abnormal increase in lysosomal membrane permeability, which has been identified as a potential factor contributing to cell death. In addition, the positively charged nanoplastics were 83% more cytotoxic than the negatively charged nanoplastics [71]. These stimuli destabilize the lysosomal membrane, leading to its rupture and hindering autophagy and cellular clearance processes, ultimately culminating in cell death. Moreover, the distribution of MNPs varied depending on the dose. At low doses, MNPs accumulated in lysosomes and lysosomal vacuoles, and at high doses, they predominantly accumulated in autolysosomes [172]. Autolysosomes, which are composed of autophagosomes and lysosomes, contain endogenous substrates such as senescent organelles or local cytoplasm [173]. The infiltration of MNPs significantly disrupts cells. Although high doses of MNPs initially triggered autophagy, prolonged exposure suppressed this process [63]. Additionally, the increased presence of autolysin due to MNP absorption inhibited the ability of the cell to eliminate them, and the exocytosis pathway was obstructed, impairing the expulsion of a large quantity of MNPs and ultimately leading to cell death [174].

### 5.3. Targeting Autophagy to Treat Diseases

Dysregulation of autophagy significantly contributes to diseases, including encompassing neurodegenerative disorders, cancers, and metabolic conditions. Currently, mutations in *ATG*, which is a core gene within the autophagy pathway, are considered primary causes of some neurodegenerative diseases including Parkinson’s disease and spinocerebellar ataxia [161,162]. Furthermore, in Alzheimer’s disease, neuronal dysfunction can be exacerbated by factors that promote amyloid formation, such as amyloid precursor proteins and presenilin, which impair lysosomal function and autophagosome clearance [175,176]. However, the inhibition of autophagy is considered advantageous in cancer treatment, because cancer cells rely more heavily on autophagy than normal cells [177,178]. Hydroxychloroquine and chloroquine can be used to treat various cancers, including multiple myeloma, glioblastoma, and melanoma by inhibiting the overall function of lysosomes and the final degradation phase of autophagy [179]. Autophagy fulfills essential metabolic roles in major organs, contributing to energy equilibrium, and complete inhibition directly impacts the composition of the extracellular metabolome, thereby affecting metabolic connections between different tissues [180]. Consequently, autophagy acts a pivotal role in responding to both short-term and long-term metabolic stresses. Alterations in autophagy influence the development of metabolic disorders like nonalcoholic fatty liver disease, type 2 diabetes, and obesity [181].

Although much work remains to be conducted, regulating autophagy for therapeutic purposes remains a promising strategy for treating a wide range of human diseases. Interventions targeting autophagy regulation also hold promise for preventing or mitigating phenotypic abnormalities in the most common human diseases.

## 6. Regulatory Mechanisms of Necroptosis Induced by MNPs and Associated Disease Risks

### 6.1. Necroptosis Caused by Micro(nano)plastics

Necroptosis is a process of PCD that relies on the activation of RIPK1, RIPK3, and MLKL and is mediated by inflammatory cell death mechanisms. When apoptosis is inhibited, both intracellular and extracellular signals may trigger necroptosis [182,183]. Previous studies have shown that MPs can upregulate the expression of RIPK3 and MLKL in mice, leading to necroptosis. Shan et al. used immortalized human brain microvascular endothelial cells, indicating that PS-NPs could be internalized by cells and induce necroptosis [74]. Similarly, Meng et al. observed necroptosis induction in the kidney following oral administration of PS-MPs to chickens [76]. Wang et al. revealed that the cytotoxicity of MPs was size-dependent. Small-sized MPs increased necroptosis in cells; PS-MPs significantly increased the protein expression levels of p-MLKL, RIPK3 and RIPK1 in MBECs and decreased caspase-8 protein expression. Treatment with N-acetyl-L-cysteine (NAC) remarkably reduced the cellular changes associated with necroptosis regulators, indicating that PS-MPs could induce necroptosis through oxidative stress [75]. Additionally, Tang et al. demonstrated that PS-NPs activated the MAPK/ROS pathway to increase lipopolysaccharide-induced necroptosis in RAW264.7 cells [184], and Wu et al. confirmed that simultaneous exposure to PS-MPs and DEHP increased necroptosis in the ovarian granulosa cells of GRM02 mice [185].

### 6.2. Regulatory Mechanisms

ROS play a crucial role in necroptosis [186]. Mitochondrial ROS production alters RIPK1 residues, facilitating RIPK1 autophosphorylation, the recruitment of RIPK3 to form amyloid-like structures, and subsequent necroptosis [187]. Additionally, MPs exacerbate intracellular ROS generation and oxidative stress after entering and accumulating within organisms, and necroptosis can be induced by stimuli such as MPs, which are known to trigger cell death [188]. RIP1 kinase is a pivotal upstream regulator in this pathway, exerting control through multiple phosphorylation and ubiquitination events. While RIPK1 is deubiquitinated and the activity of caspase-8 is inhibited, necroptosis occurs [189]. Tumor necrosis factor receptor 1 (TNFR1) can recognize the tumor necrosis factor (TNF), this process prompts the formation of membrane-associated complex I. The membrane-associated complex I is composed of cellular inhibitor of apoptosis protein 1 (cIAP1) and cIAP2, TNF-associated factor (TRAF), TNFRSF1A-associated death domain (TRADD) and receptor-interacting serine-threonine kinase 1 (RIPK1) [190]. In this complexus, RIPK1 undergoes ubiquitination by cIAP1/2 or deubiquitination by cylindromatosis protein (CYLD) [191]. RIPK1 deubiquitination leads to the dissociation of RIPK1 and TRADD from complex I, which triggers necroptosis or apoptosis [192,193]. During apoptosis, caspase-8 and FAS death domain-associated protein (FADD) are recruited to RIPK1 and TRADD, thereby activating caspase-8 to induce apoptosis [194]. Conversely, caspase-8 is absent during necroptosis, leading to RIPK1 phosphorylation and the subsequent activation of RIPK3. Activated RIPK3 phosphorylates the mixed lineage kinase domain (MLKL), inducing conformational changes and the oligomerization of MLKL [195,196]. These changes allow MLKL to disrupt the plasma membrane [197]. The resulting influx of Na^+^ and water, and efflux of K^+^, cause cell swelling, interrupt membrane potential, ultimately leading to cell death [198] (Figure 4).

### 6.3. Targeting Necroptosis to Treat Inflammation, Parkinson’s Diseases and Cancer

Necroptosis, which is a PCD pathway, significantly contributes to various human diseases due to its pivotal role in initiating inflammation [199,200]. Kang et al. reported that the RIPK1/RIPK3/MLKL/caspase-8 axis enhanced the assembly and function of the lipopolysaccharide-induced NLRP3 inflammasome in dendritic cells in the absence of caspase-8, underscoring the association of necroptosis with specific inflammatory conditions [201]. Necroptosis is an inflammatory trigger. In an in vivo mouse genetic model, Welz et al. demonstrated a link between the regulatory mechanism of necroptosis and the pathogenesis of chronic intestinal inflammation in humans [202]. Moreover, epithelial necroptosis has been observed in rectal biopsy specimens from individuals with Crohn’s disease [203]. Necroptosis is also closely associated with neurodegenerative disorders. Following spinal cord injury in mice, RIPK1, RIPK3, and MLKL have been detected in the cytoplasm and lysosomes [204]. In a Parkinson’s disease model, pharmacological inhibition of RIPK1 mitigated mitochondrial morphological alterations and dysfunction in dopaminergic neurons, indicating the beneficial impact of blocking necroptosis on Parkinson’s disease and suggesting its potential for therapeutic target [205]. In the context of cancer therapy, necroptosis holds promise as an antitumor mechanism. Given that cancer often evades apoptosis, triggering necroptosis, which is an alternative cell death pathway, is a potential treatment strategy [206]. While numerous necroptosis-targeting drugs have shown antitumor efficacy, their clinical therapeutic effects require validation [207,208,209]. Although the expression of key necroptosis pathway regulators is typically downregulated in most cancer cells, their expression is upregulated in certain cancer types [210,211]. Considering the intricate interplay between necroptosis and various diseases, further research should focus on elucidating the molecular mechanisms and characteristics of necroptosis, as well as its interactions with other cell death modalities, to identify potential cancer treatment strategies.

## 7. Regulatory Mechanisms of Apoptosis Induced by MNPs and Associated Disease Risks

### 7.1. Apoptosis Caused by Micro(nano)plastics

Apoptosis is a regulated cell death (RCD) process in which cells respond to physiological and pathological signals from the environment, alterations in environmental conditions, or injurious stimuli in an organized manner. It is morphologically characterized by cell shrinkage, the loss of cellular connections, detachment from neighboring cells, and the eventual formation of apoptotic bodies without the leakage of cellular contents, thereby preventing an inflammatory response [212,213]. Various researchers have documented that MPs can induce apoptosis in various organisms. Yang et al. demonstrated that PS-NPs induced apoptosis in N2A and JEG-3 cells through ROS production, subsequently leading to damage to GABAergic neurons in the mouse thalamus [81]. Similarly, PS-MPs exhibit phagocytic activity in human microglia HMC-3 cells, resulting in alterations in cell morphology, the immune response, and the induction of microglial apoptosis [59]. Several investigations have reported the occurrence of apoptosis in zebrafish cells following exposure to MPs. Chen et al. reported that PS-NPs exposure triggered the mitochondria-dependent apoptosis pathway in juvenile zebrafish cells [80]. Umamaheswari et al. revealed that exposure to PS-MPs upregulated the expression of *p53*, *gadd45ba* and *casp3b*, leading to apoptosis in zebrafish gills [83]. Santos et al. reported increased expression of apoptosis-related genes (*casp8*, *casp9*, and *casp3*) in zebrafish exposed to MPs [214].

### 7.2. Regulatory Mechanisms

Numerous studies have demonstrated that exposure to MNPs leads to excessive ROS production [215,216]. While ROS serve as critical cell signaling molecules, their overabundance can induce apoptosis, thereby causing tissue damage and adverse effects [217]. Among the various apoptosis pathway, mitochondria-dependent apoptosis is particularly important and has been extensively investigated [218]. Notably, mitochondria not only serve as a source of ROS but also as targets of ROS-mediated damage [219]. ROS produced by MNP exposure can upregulate the transcription of the *p53* gene, and subsequently activate *casp3b*, which promotes the transcription of gadd45ba, leading to DNA damage and apoptosis [83]. In the mitochondrial apoptosis pathway, the caspase family and Bcl-2 play indispensable roles [220]. Although only small-sized nanoparticles can be internalized by cells, larger MPs can also induce apoptosis. Exposure to MNPs triggers the generation of apoptotic signals, leading to the downregulation of the antiapoptotic gene *bcl-2* and translocation of the proapoptotic protein Bax to the mitochondrial membrane [90,221]. Consequently, mitochondrial permeability increases, resulting in the release of cytochrome c (Cyt c) into the cytoplasm, where it binds to apoptotic protease-activating factor 1 (Apaf-1) and recruits procaspase-9 to form apoptosomes [212]. During apoptosis, caspase-9 undergoes self-proteolytic cleavage, which initiates the caspase processing cascade (Figure 5). Moreover, MPs influence the concentration of calcium ions (Ca^2+^) in mitochondria by inducing changes in mitochondrial membrane potential [222], thereby impairing adenosine triphosphate (ATP) production, activating downstream caspase-9 and caspase-3, and ultimately triggering the mitochondrial apoptosis pathway [223].

### 7.3. Targeting Apoptosis to Treat Lung Diseases, Neurodegenerative Diseases, Cancer and Other Diseases

Apoptosis plays a crucial role in chronic obstructive pulmonary disease (COPD), and there is a notable increase in lung apoptosis that is often attributed to cigarette smoke exposure [224]. Animal experiments further showed the involvement of apoptosis in COPD pathogenesis. Mice exposed to cigarette smoke exhibited lung cell apoptosis and subsequently developed emphysema after tracheal perfusion of caspase. Conversely, mice treated with Caspase-3 inhibitors or subjected to Caspase gene deletion showed reduced lung damage [225,226]. As a neurodegenerative disease that may lead to brain death, stroke is also associated with apoptosis. Given its energy-dependent nature, apoptosis is notably prevalent in neurons in the ischemic penumbra with compromised energy production [227]. Studies have shown that organisms can influence pathways such as LRRK2 and GSK-3 beta/Tau hyperphosphorylation/Nrf2, as well as the aggregation of alpha-synuclein, through apoptotic cell death [228,229]. The deletion of a single proapoptotic *Bax* gene plays a neuroprotective role [230]. Too little apoptosis is one of the pathogenic factors associated with cancer because a reduction in apoptosis can allow malignant cells to continue to survive [231]. Although apoptosis is one of the causes of cancer, it is also a potential target for cancer treatment. Recent research has identified various effective substances that can inhibit apoptosis. Chen et al. reported that exogenous NO significantly mitigated NP-induced apoptosis by targeting mitochondria [80], and Yang et al. discovered that FGFR4 and EZH2 inhibitors synergistically induced HCC cell apoptosis by inhibiting YAP signaling [232]. The potential of targeting apoptosis to treat diseases is substantial, indicating promising avenues for future clinical research.

## 8. Cuproptosis

In addition to classic PCD, a recent research by Tsvetkov et al. revealed that copper inside cells induced a novel form of regulated cell death that was unlike oxidative stress-related cell death: cuproptosis [233]. Three mechanisms are known to trigger cuproptosis. First, FDX1 reduces Cu^2+^ to Cu^+^ and promotes the lipylation (LA) and accumulation of enzymes involved in the regulation of the mitochondrial TCA cycle, especially DLAT. FDX1 causes an imbalance in Fe-S cluster proteins to trigger cuproptosis [234]. Second, excess copper induces cell death through the oligomerization of mitochondrial lipidated proteins [235]. Although copper is essential for cellular metabolism, excess copper can be deleterious. The third mechanism involves inhibiting enzymes that are necessary for apoptosis. Copper acts as both an endoplasmic reticulum (ER) stress inducer and a caspase-3 inhibitor, forcing cells to undergo caspase-independent cell death [236,237]. Several studies have shown that MPs can be used as carriers for adsorbing Cu^2+^ [238], and MPs exposed to coastal seawater can form biofilms, thereby promoting the adsorption and transport of Cu^2+^. Studies have shown that exposure to microplastics and Cu^2+^ can increase the accumulation of Cu^2+^ in the body. Qiao’s study compared zebrafish exposed to PS-MPs and Cu^2+^ with those exposed to Cu^2+^ alone and revealed an increase in the toxic effect of MPs on Cu^2+^ [239]. However, whether cuproptosis occurs in these contexts warrants further investigation.

Copper is implicated in certain diseases, notably Wilson’s disease, which is an autosomal recessive genetic disease typical of various mutations in the *ATP7B* gene. *ATP7B* dysfunction impairs copper excretion, resulting in the accumulation of copper in the liver and brain. Cuproptosis has emerged as a potential therapeutic target for Wilson’s disease, and copper chelators such as trientine and D-penicillamine are considered effective treatments [240]. Furthermore, Liu et al. suggested that targeting cuproptosis could serve as a potential therapeutic approach for kidney cancer and brain cancer, based on pancancer analysis of genomic and transcriptomic data associated with cuproptosis [241].

## 9. Conclusions

This review highlighted the adverse effects of MNPs on organisms and elucidated the various mechanisms leading to cell death and resulting in a threat to normal organismal development. Numerous researchers have demonstrated the detrimental effects of MNPs on various species, including humans, fish, mice, chickens, and nematodes, which are characterized by tissue damage, neurotoxicity, developmental toxicity, and oxidative stress. These data indicate that MNPs in in vivo studies are typically exposed to low doses over prolonged periods, whereas in vitro studies employ higher exposure concentrations but for shorter durations. Most research predominantly examines the impact of PS on organisms, neglecting the effects of other plastics such as PE, PP, and PVC. Moreover, the heightened toxicity of MNPs modified by functional groups (-NH_3_, -COOH and -SO_3_H) warrants attention, as they often exhibit greater toxicity than unmodified MNPs. Additionally, environmental endocrine disruptor chemicals (EDCs) capable of interacting with MNPs in the environment should be a concern. Given the detection of MNPs in human placentas, genetic variability has emerged as a significant factor influencing the impact of MNPs on cellular health. Consequently, there is a pressing need for further research on microplastic pollution to comprehensively assess the interaction between microplastics and human health. Genetic and pathological investigations have provided compelling evidence linking genetic mutations in various forms of PCD (including pyroptosis, ferroptosis, autophagy, necroptosis, apoptosis, and cuproptosis) and the onset of numerous human diseases, underscoring the significance of programmed death and associated cellular functions in disease pathogenesis. However, due to limited direct evidence regarding the health risks posed by MNP exposure to humans, significant knowledge gaps persist concerning the fate of MNPs and their adverse influence on human health. Currently, research about the impact of MNPs on human health has predominantly been confined to in vitro studies, indicating that investigations are still in the nascent stages. Therefore, there is an urgent need to perform systematic and comprehensive studies to determine the actual human exposure concentrations, distribution patterns, metabolic pathways and the size and concentration of toxic MNPs in humans and to develop a standard, if possible, to elucidate their effects on human health.

## Figures and Tables

**Figure 1 toxics-12-00493-f001:**
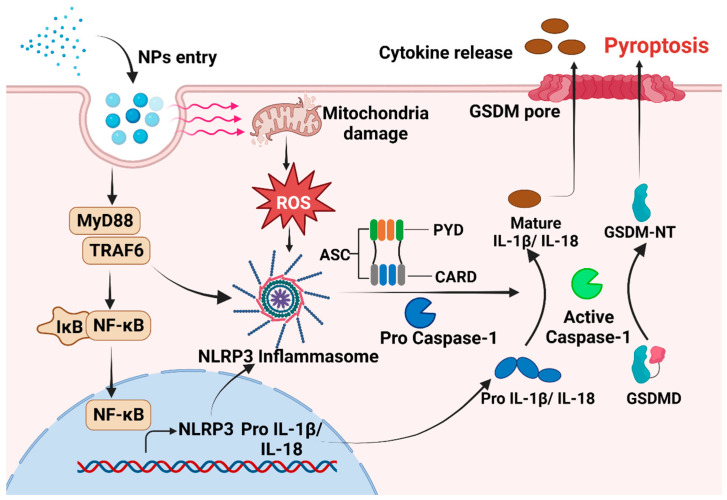
The mechanism underlying pyroptosis induced by NPs. After NPs enter the cell, there is an increase in intracellular ROS levels. This increase in ROS activates NLRP3, which then binds to procaspase via the adaptor protein ASC. In addition, MNPs can trigger the inflammatory NF-κB /MyD88/NLRP3 pathway. Subsequently, activated Caspase-1 is formed, leading to the initiation of pyroptosis.

**Figure 2 toxics-12-00493-f002:**
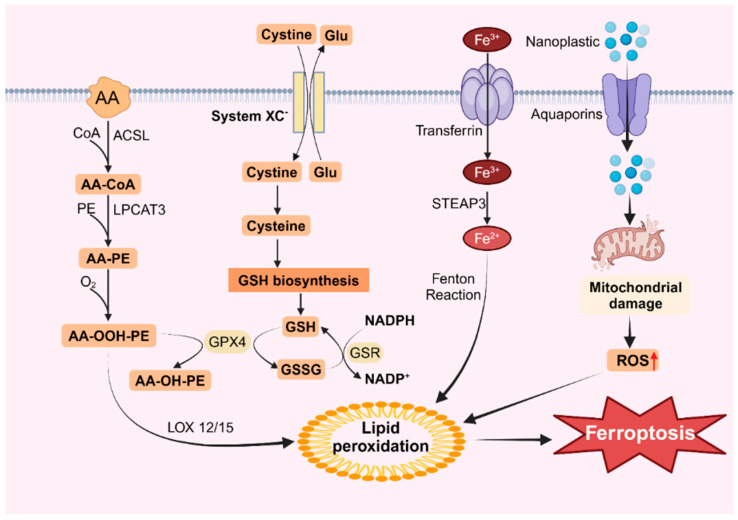
Ferroptosis caused by nanoplastics. Nanoplastics can enter cells through aquaporins, damage mitochondria and increase ROS levels, leading to lipid peroxidation and inducing ferroptosis.

**Figure 3 toxics-12-00493-f003:**
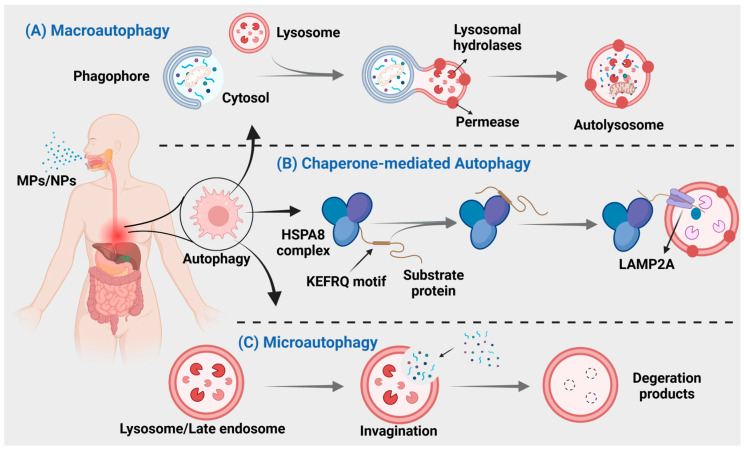
MNPs can cause three different types of autophagy. (**A**) Autophagosomes derived from the endoplasmic reticulum, Golgi apparatus, or plasma membrane, engulf the material and subsequently merge with lysosomes for degradation. (**B**) Direct encapsulation of long-lived proteins by lysosomal membranes, leading to their degradation within lysosomes. (**C**) Intracytoplasmic proteins bind to chaperones for transport into lysosomal cavities, followed by digestion by lysosomal enzymes.

**Figure 4 toxics-12-00493-f004:**
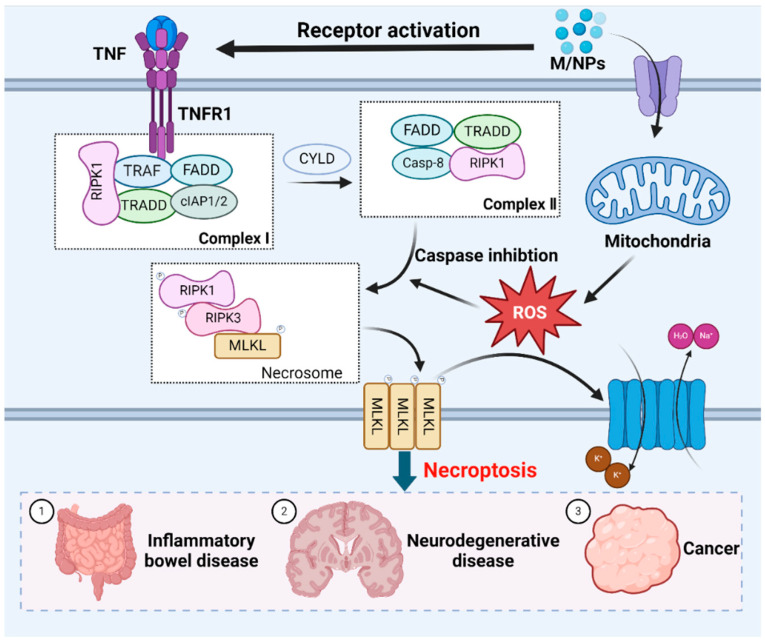
The molecular mechanisms underlying necroptotic cell death induced by MNPs. Exposure to MNPs triggers necroptosis mediated by RIPK1, which requires RIPK3-dependent phosphorylation of MLKL. Oligomerization of MLKL culminates in plasma membrane disruption, which is accompanied by the efflux K^+^, and influx of Na^+^ and H_2_O, resulting in cell swelling. Necroptosis is closely related to inflammatory bowel disease, neurodegenerative disease and cancer.

**Figure 5 toxics-12-00493-f005:**
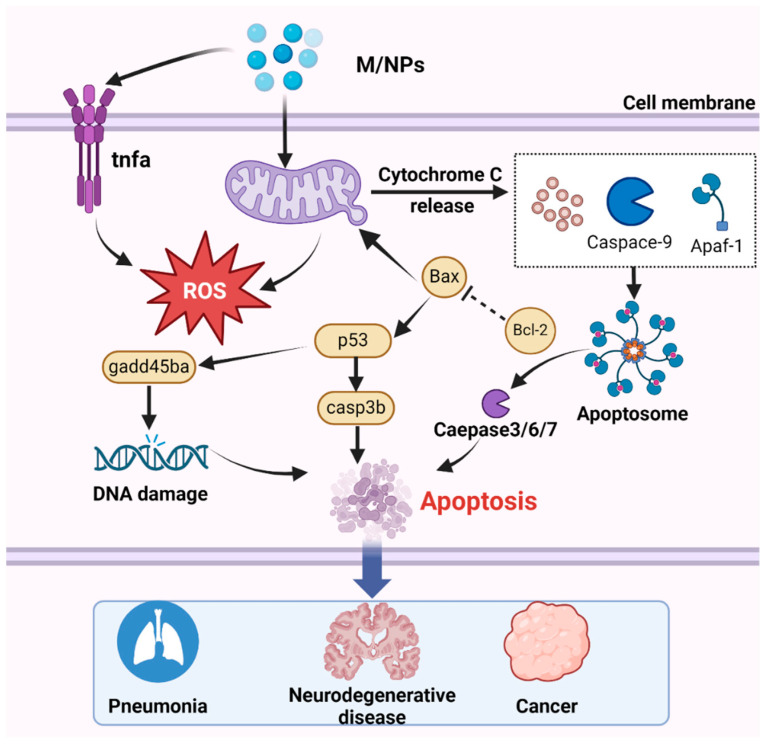
The mechanism by which MNPs induce apoptosis involves several key steps. After entering the cell, with the accumulation of MNPs, MNPs trigger mitochondrial detoxification, leading to the generation of ROS. Subsequently, the increase in ROS levels upregulates the transcription of the *p53* gene, thereby activating casp3b. Activated caspase-3 promotes the transcription of gadd45ba, ultimately resulting in DNA damage and apoptosis. Additionally, Cyt c is released by the mitochondria, binds to Apaf-1, and facilitates the recruitment of procaspase-9 and the formation of apoptotic bodies, then activated effector caspase, thus inducing apoptosis. Apoptosis is associated with numerous diseases, including pneumonia, neurodegenerative disorders, and cancer.

**Table 1 toxics-12-00493-t001:** Cell death caused by MNPs, sorted by the type of PCD.

Model System	Particle Type/Size	Exposure Time	Exposure Dose	Types of Cell Death	Ref.
Mice, male, 5 weeks old	PP/5 μm	28 d	0.1, 0.5, and 1 mg/mL	Pyroptosis and ferroptosis	[51]
Wistar rats, 6 weeks old	PS/0.5 μm	90 d	0.5, 5, and 50 mg/L	Pyroptosis and apoptosis	[52]
Wistar rats, male, 6 weeks old	PS/0.5 μm	90 d	0.015, 0.15, and 1.5 mg/kg	Pyroptosis and apoptosis	[53]
Chicks, 1 day old	PS/5 μm	42 d	1, 10, and 100 mg/L	Pyroptosis	[54]
Mouse microglial cell line (BV2)	PS/44 nm	12 h or 24 h	0, 25, 50, or 100 µg/mL	Ferroptosis	[55]
Chickens, 1 day old	PS/5 μm	42 d	1, 10, and 100 mg/L	Ferroptosis and apoptosis	[56]
C57BL/6J mice, 6−8 weeks old	PS/80 nm	Three times per week during pregnancy	1, 5, and 25 μg/μL	Ferroptosis	[57]
SPF C57BL/6 mice, male, 6 weeks old	PS/140.63 ± 8.96 nm	28 d	0.25 mg/kg	Autophagy	[58]
ICR mice, 8 weeks old	PS/60 or 900 nm	Ingested during the 9.5 or 15th day of pregnancy	300 μg/time	Autophagy and apoptosis	[59]
HUVECs	PS/100 and 500 nm	48 h	25 μg/mL	Autophagy	[60]
HNEpCs	PS/50 and 500 nm	24 h	100 μg/mL	Autophagy	[61]
GES-1 cells	PS/60 nm	12, 24, 48 h	50 μg/mL	Apoptosis and autophagy	[62]
Human Kidney Proximal Tubular Epithelial Cells HK-2	PS/2 μm	24 or 48 h	0.05, 0.1, 0.2, 0.4, or 0:8 mg/mL	Autophagy and apoptosis	[63]
BEAS-2B human bronchial epithelial cells	PS/99.4 nm	24 h	25, 50, 100, 200 μg/mL	Autophagy	[64]
High land broilers	PS/5 μm	42 d	1, 10, and 100 mg/L	Autophagy and apoptosis	[65]
HEK293 cells	PS/3.54 ± 0.39 μm	24 h	300 ng/mL	Autophagy and apoptosis	[66]
Zebrafish (*Danio rerio*); freshwater perch (*Perca fluviatilis*)	PP/8–10 µm	21 d	1 and 10 mg/g	Autophagy and apoptosis	[67]
ZF4 cells	PS/1000 nm	1, 3, 6, 9 h	20 μg/mL	Autophagy and apoptosis	[68]
*Caenorhabditis elegans*	PS/100 nm	5 d	100 mg/L	Autophagy	[69]
HaCaT cells	PE/30–300 nm	48 h	10, 50, 100 μg/mL	Autophagy	[70]
Earthworm (*Eisenia fetida*)	PS/69.7–197.7 nm	42 d	0.3, 3 mg/kg	Autophagy	[71]
Chicks, one day old	PS/5 μm	42 d	1, 10, and 100 mg/L	Autophagy	[72]
Caco-2 cells	PS/100 nm	24 h	120 μg/mL	Necroptosis	[73]
hCMEC	PS/50 nm	72 h	100 μg/mL	Necroptosis	[74]
C57BL/6 mice, male, 6 weeks old	PS/1–10 μm, 5–100 μm	30 d	10 mg/L	Necroptosis	[75]
Chicken, 1 day old	PS/5 μm	42 d	1, 10, 100 mg/L	Necroptosis	[76]
Swine testis cells	PS/1–10 μm	24 h	250, 500, and 1000 μg/mL	Apoptosis and necroptosis	[77]
Caco-2 cells	PS/100 nm	48 h	30 μg/mL	Apoptosis	[78]
Carp	PS/50, 100, 400 nm	28 d	1000 μg/L	Apoptosis	[79]
Zebrafish (*Danio rerio*), larvae.	PS/50 nm	5, 7.5, 10 d	20 mg/L	Apoptosis and ferroptosis	[80]
SPF C57BL mice, 8 weeks old	PS/100 nm	17 d	1 mg/d	Apoptosis	[81]
Human microglial HMC-3 cells	PS/0.2, 2 μm	24 h	10 μg/mL	Apoptosis	[82]
Zebrafish (*Danio rerio*), male, adult	PS/0.1–0.12 μm	35 d	10, 100 μg/mL	Apoptosis	[83]
Human alveolar epithelial A549 cell line	PS/25, 70 nm	2, 4, 8 h	25, 160 μg/mL	Apoptosis	[84]
Mice, 6 weeks old	PS/50, 500 and 5000 nm	28 d	2.5–500 mg/kg body weight	Apoptosis	[85]

PP, polypropylene; PS, polystyrene; ICR, Institute of Cancer Research; SPF, specific pathogen-free; HUVECs, human umbilical vein endothelial cells; HNEpCs, human nasal epithelial cells; HaCaT, human keratinocytes; hCMECs, human cerebral microvascular endothelial cells.

**Table 2 toxics-12-00493-t002:** The link between ferroptosis and disease.

Disease Type	Species	Target Protein	Therapeutic	Ref.
Cardiomyopathy	Mice	Hmox1	Dexrazoxane, Fer-1	[152]
Intestinal I/R injury	Mice, Caco-2 cell	ACSL4	Liproxstatin-1	[153]
Colorectal cancer	Colorectal cancer stem cells	SLC7A11	Erastin	[154]
Ulcerative colitis	HCoEpiC cell, Mice	NF-κBp65	GSK 414	[155]
Heart injury induced by Ischemia reperfusion	Mouse embryonic fibroblasts	Enzymes involved in glutaminolysis	Iron chelator and a glutaminase-2 inhibitor	[148]
Cancer	Ferroptosis-resistant MCF7 cells	CoQ10	FSP1	[156]
Liver cancer	Hepatocellular carcinoma cells (Huh7, Hep3B, HepG2)	Sigma1 receptor, metallothionein-1	DFX, sorafenib	[157]
Pancreatic cancer	PANC1, CFPAC1, MiaPaCa2, Panc2.03, and Panc02 cells	HSPA5 and GPX4	Sulfasalazine, Epigallocatechin	[158]
Parkinson’s disease	Humans	α-syn	Deferiprone	[159]
Alzheimer’s disease	Mice, Human cerebral cortex tissue	Fpn	Liproxstatin1, ferrostatin1	[160]

## Data Availability

Not applicable.

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
