# Peer review of "Research Progress on Micro(nano)plastic-Induced Programmed Cell Death Associated with Disease Risks"

_toxics, 2024, doi:10.3390/toxics12070493_

Round 1
Reviewer 1 Report
Comments and Suggestions for Authors
Overall, this review is focused mainly on the MNPs induced programmed cell death and associated risks to certain diseases, which is reflected in the title. It is well written, however discussion about potential treatment strategies is very general and hypothetical. Even though it is a nice discussion, this can be done at the stage when more information is available. I would leave this on discretion of the main editor however I recommend excluding discussion concerning suggestion of treatment strategies, they seem to be out of place. The title of the review does not mention any potential treatment strategies. After following changes are made, the review can be accepted for publication.
Abstract: Following sentence (Lines 11-12, food chains through multiple pathways?) it should be reworded, as it reads now, it does not reflect inhalation exposure to MNPs, which is also a crucial route of exposure.
Table 1 is comprehensive, but it doesn’t seem to be sorted, if it is sorted certain way, it wasn’t easily noticeable. I recommend organizing the papers in this Table by either the type of PCD, by the type of MNPs or type of animal model/cell type for easier orientation, they can be groups into these categories and the title of the table should mention how they are being sorted.
Under definition of Microplastics, it is mentioned that they are irregular plastic fragments (which might not always be true), even regular spherical plastic particle under 5 mm would fall under category of microplastics, thus “irregular” needs to be removed (line 29).
When NPs or MPs are mentioned in the text, referring to previous work, it should also be stated what type of plastics they were (chemical composition at least, as listed in Table 1).
Graphical abstract: change “nerve disease” to “nervous system diseases”.
Minor corrections that should be done:
Line 151: remove “will”.
Line 157: the sentence is not finished.
Author Response
Dear editor and reviewers:
We thank you for your careful screening of the manuscript and insightful comments. In what follows, we have responded to the requests for clarification in detail point-by-point. In addition, we tried our best to check and revise our manuscript to improve the quality of our manuscript. The changed content was highlighted in red font in the revised manuscript. We hope that your concerns have been fully addressed. The following are my responses to the reviewers’ comments one by one.
Comment 1: Abstract: Following sentence (Lines 11-12, food chains through multiple pathways?) it should be reworded, as it reads now, it does not reflect inhalation exposure to MNPs, which is also a crucial route of exposure.
Response to Comment 1: Thanks for your kindly suggestion. We added two other ingestion pathways of MNPs, including inhalation and dermal contact between line 11 to 12 in the revised manuscript.
The modifications are as follows:
They infiltrate various organisms within different food chains through multiple pathways including inhalation and dermal contact, and pose a significant environmental challenge in the 21st century.
Comment 2: Table 1 is comprehensive, but it doesn’t seem to be sorted, if it is sorted certain way, it wasn’t easily noticeable. I recommend organizing the papers in this Table by either the type of PCD, by the type of MNPs or type of animal model/cell type for easier orientation, they can be groups into these categories and the title of the table should mention how they are being sorted.
Response to Comment 2: Thank you very much for your advice. We sorted Table 1 by the type of PCD and add the title of the table, mentioned how they were being sorted.
Comment 3: Under definition of Microplastics, it is mentioned that they are irregular plastic fragments (which might not always be true), even regular spherical plastic particle under 5 mm would fall under category of microplastics, thus “irregular” needs to be removed (line 29).
Response to Comment 3: Thank you for your nice advice We removed “irregular” in line 30 in the revised manuscript.
The modifications are as follows:
Microplastics (MPs) are plastic fragments and particles with diameters less than 5 mm, and particles smaller than 1000 nm are classified as nanoplastics (NPs).
Comment 4: When NPs or MPs are mentioned in the text, referring to previous work, it should also be stated what type of plastics they were (chemical composition at least, as listed in Table 1).
Response to Comment 4: Thanks very much for your suggestion. We stated what type of plastics they were in the revised manuscript, for example in line 51, 57, 251, 256, ect.
Comment 5: Graphical abstract: change “nerve disease” to “nervous system diseases”.
Response to Comment 5: Thank you for your excellent suggestion. We changed “nerve disease” to “nervous system diseases” in graphical abstract.
The modifications are as follows:

Comment 6: Minor corrections that should be done:
Line 151: remove “will”.
Line 157: the sentence is not finished.
Response to Comment 6: Thanks for your suggestion. We removed “will” in line 165 and finish the sentence between line 169 to 171 in the revised manuscript.
The modifications are as follows:
These findings suggest that MPs are a potential cause of some cardiovascular diseases, such as myocarditis, atherosclerosis and myocardial infarction.
Exposure to MNPs can result in the excessive accumulation of reactive oxygen species (ROS) within organisms and imbalance between oxidation and antioxidant effects, consequently inducing oxidative stress.

Reviewer 2 Report
Comments and Suggestions for Authors
This review provides a detailed summary of recent literature that has examined the potential role of micro/nanoplastics (MNPs) in initiating programmed cell death by several different mechanisms and of the potential impact on the development of disease or, in some cases, on development of therapeutic approaches. The draft provides a large amount of useful information on potential effects of this wide-spread environmental contaminant. However, there are some questions that should be addressed to increase the potential impact of the review.
1. Table 1 summarizes 34 papers in which MNPs have been shown to lead to cell death by specific mechanisms. It is not clear to me what the purpose of this table is, since the later sections on specific types of programmed cell death do not appear to cover these references. It is also quite striking that all but one study in Table 1 has looked at polystyrene, but there is no mention of that until the Conclusions where it is noted that most research has used polystyrene. To summarize, more discussion of Table 1 is needed: why is it included and what can one conclude?
2. Table 1 is useful as it specifies the type of MNP, and gives some information on the model organism/cell type, particle size, and exposure dose and time. This information is of significant importance if one is to understand the potential biological impact of MNPs and the review would benefit from a more in-depth description or summary of these factors in the various studies that are covered later. I guess that some information on the importance of physical chemical properties and the exposure model is in reference 49, but a short summary could be included here to provide context for the information in this review.
3. It would be very beneficial to provide a summary of what is known about the prevalence of the various possible types of MNPs that are present in environmental compartments or to which humans are exposed via various routes. Furthermore, are there studies that show MNP types or concentrations or exposure routes for which no biological effects are observed?
4. The first two paragraphs of section 2 (and Figure S2) outline the approach taken to identify the references that are summarized here. It would be helpful to provide the number of papers that deal specifically with MNPs, as well as the total number of papers. It would also be helpful to understand the reasons (beyond a sentence that comments on “impact”) for which ~160 of the papers initially identified were excluded.
5. Table 2 highlights references that indicate a possible link between ferroptosis and disease. It seems that some of the other PCD sections would also benefit from a table that summarizes some of the data.
6. There are a lot of abbreviations in the review and a lack of consistency in their use. One example that stands out is the use of polystyrene vs PS, both of which are used throughout. PCD vs programmed cell death is another example. The authors should check carefully that when an abbreviation is defined, it is used consistently throughout.
7. There are quite a few references where the author names are missing (only the first initials provided), eg, references 4, 5, 6. The problem is more prevalent in the first 50 or so references but there are instances throughout.
Comments on the Quality of English Language
This review provides a detailed summary of recent literature that has examined the potential role of micro/nanoplastics (MNPs) in initiating programmed cell death by several different mechanisms and of the potential impact on the development of disease or, in some cases, on development of therapeutic approaches. The draft provides a large amount of useful information on potential effects of this wide-spread environmental contaminant. However, there are some questions that should be addressed to increase the potential impact of the review.
1. Table 1 summarizes 34 papers in which MNPs have been shown to lead to cell death by specific mechanisms. It is not clear to me what the purpose of this table is, since the later sections on specific types of programmed cell death do not appear to cover these references. It is also quite striking that all but one study in Table 1 has looked at polystyrene, but there is no mention of that until the Conclusions where it is noted that most research has used polystyrene. To summarize, more discussion of Table 1 is needed: why is it included and what can one conclude?
2. Table 1 is useful as it specifies the type of MNP, and gives some information on the model organism/cell type, particle size, and exposure dose and time. This information is of significant importance if one is to understand the potential biological impact of MNPs and the review would benefit from a more in-depth description or summary of these factors in the various studies that are covered later. I guess that some information on the importance of physical chemical properties and the exposure model is in reference 49, but a short summary could be included here to provide context for the information in this review.
3. It would be very beneficial to provide a summary of what is known about the prevalence of the various possible types of MNPs that are present in environmental compartments or to which humans are exposed via various routes. Furthermore, are there studies that show MNP types or concentrations or exposure routes for which no biological effects are observed?
4. The first two paragraphs of section 2 (and Figure S2) outline the approach taken to identify the references that are summarized here. It would be helpful to provide the number of papers that deal specifically with MNPs, as well as the total number of papers. It would also be helpful to understand the reasons (beyond a sentence that comments on “impact”) for which ~160 of the papers initially identified were excluded.
5. Table 2 highlights references that indicate a possible link between ferroptosis and disease. It seems that some of the other PCD sections would also benefit from a table that summarizes some of the data.
6. There are a lot of abbreviations in the review and a lack of consistency in their use. One example that stands out is the use of polystyrene vs PS, both of which are used throughout. PCD vs programmed cell death is another example. The authors should check carefully that when an abbreviation is defined, it is used consistently throughout.
7. There are quite a few references where the author names are missing (only the first initials provided), eg, references 4, 5, 6. The problem is more prevalent in the first 50 or so references but there are instances throughout.
Author Response
Dear editor and reviewers:
We thank you for your careful screening of the manuscript and insightful comments. In what follows, we have responded to the requests for clarification in detail point-by-point. In addition, we tried our best to check and revise our manuscript to improve the quality of our manuscript. The changed content was highlighted in red font in the revised manuscript. We hope that your concerns have been fully addressed. The following are my responses to the reviewers’ comments one by one.
Comment 1: Table 1 summarizes 34 papers in which MNPs have been shown to lead to cell death by specific mechanisms. It is not clear to me what the purpose of this table is, since the later sections on specific types of programmed cell death do not appear to cover these references. It is also quite striking that all but one study in Table 1 has looked at polystyrene, but there is no mention of that until the Conclusions where it is noted that most research has used polystyrene. To summarize, more discussion of Table 1 is needed: why is it included and what can one conclude?
Response to Comment 1: Thanks for your kindly suggestion. We explained why these 34 papers were included and drawn some conclusions between line 102 to 107 in the revised manuscript.
The modifications are as follows:
To understand the potential biological impact of MNPs, we collected nearly five years of research papers on PCD caused by MNPs and were surprised to find that only two studies used PP-MPs, while the other studies used PS-MNPs. And long-term exposure at low doses is usually studied in animals, whereas acute exposure at higher doses is typically examined in cells.
Comment 2: Table 1 is useful as it specifies the type of MNP, and gives some information on the model organism/cell type, particle size, and exposure dose and time. This information is of significant importance if one is to understand the potential biological impact of MNPs and the review would benefit from a more in-depth description or summary of these factors in the various studies that are covered later. I guess that some information on the importance of physical chemical properties and the exposure model is in reference 49, but a short summary could be included here to provide context for the information in this review.
Response to Comment 2: Thank you very much for your advice. We added a short summary to describe the importance of physical chemical properties and the exposure model between line 98 to 102 in the revised manuscript.
The modifications are as follows:
The particle size of MNPs, differences in surface-modified charge, aging, and co-exposure with other substances often influence changes in the mode of PCD. Similarly, different exposure models produce varying toxic effects. Model animals, such as mice, zebrafish, and chickens, as well as specific cell lines, are often used to investigate MNP-induced PCD.
Comment 3: It would be very beneficial to provide a summary of what is known about the prevalence of the various possible types of MNPs that are present in environmental compartments or to which humans are exposed via various routes. Furthermore, are there studies that show MNP types or concentrations or exposure routes for which no biological effects are observed?
Response to Comment 3: Thank you for your nice advice. We added the type distribution of MNPs in groundwater environment between line 42 to 45 in the revised manuscript. Furthermore, while the studies we used all showed biological effects after exposure to MNPs, this is because the exposure was long enough and the exposure concentration was high enough. Because MNPs is sublethal, it is often difficult for MNPs to produce biological effects when the concentration is low enough or the exposure time is short enough.
The modifications are as follows:
Bharath K et al. examined the type of MNPs in groundwater and found that the main components were PS and PP, small amounts of polyethylene (PE) and polyvinyl chlo-ride (PVC) were also found.
Comment 4: The first two paragraphs of section 2 (and Figure S2) outline the approach taken to identify the references that are summarized here. It would be helpful to provide the number of papers that deal specifically with MNPs, as well as the total number of papers. It would also be helpful to understand the reasons (beyond a sentence that comments on “impact”) for which ~160 of the papers initially identified were excluded.
Response to Comment 4: Thank you for your suggestion. We provide the number of papers that deal specifically with MNPs and explained why the papers were excluded between line 134 to 136 in the revised manuscript.
The modifications are as follows:
Initially, 423 papers were preliminarily identified over the past decade, among them, there are 320 studies on MNPs, of which 121 studies focus on the generation and analysis of MNPs, and pay little attention to the biological effects of MNPs.
Comment 5: Table 2 highlights references that indicate a possible link between ferroptosis and disease. It seems that some of the other PCD sections would also benefit from a table that summarizes some of the data.
Response to Comment 5: Thanks very much for your suggestion. We have carefully considered your suggestion, but listing every link between programmed cell death and disease in tables is too informative and may result in too long a space to highlight the topic of this article.
Comment 6: There are a lot of abbreviations in the review and a lack of consistency in their use. One example that stands out is the use of polystyrene vs PS, both of which are used throughout. PCD vs programmed cell death is another example. The authors should check carefully that when an abbreviation is defined, it is used consistently throughout.
Response to Comment 6: Thank you for your nice advice. We carefully checked the use of abbreviations in the revised manuscript.
Comment 7: There are quite a few references where the author names are missing (only the first initials provided), eg, references 4, 5, 6. The problem is more prevalent in the first 50 or so references but there are instances throughout.
Response to Comment 7: Thank you for your excellent suggestion. We checked the references carefully and added the names of the authors in the revised manuscript.
Comment 8: This review provides a detailed summary of recent literature that has examined the potential role of micro/nanoplastics (MNPs) in initiating programmed cell death by several different mechanisms and of the potential impact on the development of disease or, in some cases, on development of therapeutic approaches. The draft provides a large amount of useful information on potential effects of this wide-spread environmental contaminant. However, there are some questions that should be addressed to increase the potential impact of the review.
Response to Comment 8: Thanks for your kindly suggestion. Thanks for your suggestion, the language of this manuscript had been edited on the website "https://secure.authorservices.springernature.com/login?redirect=/researcher/index/details/XT9G8HJZ/thank-you", and we had tried our best to eliminate punctuation and grammar mistakes in the revised manuscript (note: edit credential as follow). If further refinement is still needed, we can revise it again.
